# Development of a Mobile Device for Odor Identification and Optimization of Its Measurement Protocol Based on the Free-Hand Measurement

**DOI:** 10.3390/s20216190

**Published:** 2020-10-30

**Authors:** Gaku Imamura, Genki Yoshikawa

**Affiliations:** 1International Center for Materials Nanoarchitectonics (MANA), National Institute for Materials Science (NIMS), World Premier International Research Center Initiative (WPI), 1-1 Namiki, Tsukuba 305-0044, Japan; 2Center for Functional Sensor & Actuator (CFSN), Research Center for Functional Materials, National Institute for Materials Science (NIMS), 1-1 Namiki, Tsukuba 305-0044, Japan; YOSHIKAWA.Genki@nims.go.jp; 3Materials Science and Engineering, Graduate School of Pure and Applied Science, University of Tsukuba, 1-1-1 Tennodai, Tsukuba 305-8571, Japan

**Keywords:** machine olfaction, Membrane-type Surface stress Sensor (MSS), Transfer Function Ratio (TFR), free-hand measurement

## Abstract

Practical applications of machine olfaction have been eagerly awaited. A free-hand measurement, in which a measurement device is manually exposed to sample odors, is expected to be a key technology to realize practical machine olfaction. To implement odor identification systems based on the free-hand measurement, the comprehensive development of a measurement system including hardware, measurement protocols, and data analysis is necessary. In this study, we developed palm-size wireless odor measurement devices equipped with Membrane-type Surface stress Sensors (MSS) and investigated the effect of measurement protocols and feature selection on odor identification. By using the device, we measured vapors of liquids as odor samples through the free-hand measurement in different protocols. From the measurement data obtained with these protocols, datasets of transfer function ratios (TFRs) were created and analyzed by clustering and machine learning classification. It has been revealed that TFRs in the low-frequency range below 1 Hz notably contributed to vapor identification because the frequency components in that range reflect the dynamics of the detection mechanism of MSS. We also showed the optimal measurement protocol for accurate classification. This study has shown a guideline of the free-hand measurement and will contribute to the practical implementation of machine olfaction in society.

## 1. Introduction

Among the five human senses (i.e., sight (vision), hearing (audition), touch (somatosensation), taste (gustation), and smell (olfaction)), machine olfaction and gustation applications have not been widely implemented in society. Compared to gustation, it is more challenging to realize machine olfaction because there are no “basic smells” for olfaction whereas taste can be sensed by detecting the five basic tastes (i.e., sweet, sour, salty, umami, and bitter) [1,2]. As first demonstrated by Persaud et al. in 1982 [3], machine olfaction—gas sensor systems for identifying odors—has been extensively studied by many researchers [4,5]. Owing to the advancement in sensing technologies including the development of sensor elements, fabrication techniques, and signal analysis methods, some products have already been available in the market. However, for practical application, there are still many technical issues in sensitivity, selectivity, usability, commercial availability, and so on. Considering the recent evolution of the Internet of Things (IoT), the realization of practical machine olfaction is eagerly awaited in various fields such as food, cosmetics, housing, and medical fields [6].

One of the issues for practical applications of machine olfaction is the development of mobile measurement devices. Although the miniaturization of the sensor elements has been achieved owing to the advancement in the microelectromechanical systems (MEMS) technology [7,8], it is still challenging to make a small measurement device that includes whole measurement elements. This is mainly due to gas flow lines including pumps; gas flow rates of odor samples need to be strictly controlled in many cases because signal features characteristic to odors are difficult to be extracted without precise gas flow control [9]. For such reasons, current odor measurement devices tend to be bulky and power-consuming. To resolve this issue, the development of new sensing methods without such gas flow control is required. Nimsuk et al. developed gas classification systems for gas flow in which the concentration of samples dynamically changed. They focused on the frequency components of the sensing signals and demonstrated gas classification based on them [10,11]. Trincavelli et al. developed gas discrimination robot systems in an open-sampling condition, in which the gas flow of sample to sensors was not controlled [12,13]. In their system, gases were identified by focusing on the transient responses of segmented sensing signals obtained with metal oxide (MOX) sensors. Vergara et al. also developed a MOX sensor-based gas identification system in an open-sampling condition. The system detected the turbulent plume of sample gases in a wind tunnel facility and identified the gas species by the static responses of arrayed MOX sensors [14].

In 2019, a new measurement protocol was proposed based on transfer function ratios (TFRs), which are intrinsically independent of gas flow control and can be calculated only from sensing signals [15]. On the basis of data analysis methods using TFRs, we demonstrated odor identification through “free-hand measurement,” in which a small sensor array is manually brought close to target odors. The free-hand measurement could be achieved owing to the novel characteristics of Membrane-type Surface stress Sensors (MSS) [16]. MSS are small sensors fabricated through a MEMS process [17,18]. MSS have high sensitivity and high chemical versatility, which are advantageous to the sensor element of machine olfaction. Another merit of using MSS is its relatively high resistance against mechanical vibration and shock. Compared to gas sensors that detect shifts in resonant frequencies caused by gas sorption (e.g., quartz crystal microbalance, microcantilever-type sensors in the dynamic mode [19,20,21]), sensing responses of MSS are rather robust against physical shocks like touching or moving measurement devices owing to a full-Wheatstone bridge configuration with a relatively small mass of the sensing part. The small footprint of MSS also enables it to configure a small array of sensor channels, which can be considered to be spatially equivalent for the gas input, satisfying the intrinsic requirement to calculate TFRs. Thus, the combination of MSS and the data analysis method based on TFRs makes it possible to identify odors through the free-hand measurements. There is, however, still room for improving the protocol of the free-hand measurement. As the free-hand measurement is based on the data analysis methods using TFRs, the input pattern of the protocol—the motion of a hand holding a measurement device—should contain a wide range of frequency components. For that reason, an ideal free-hand measurement protocol should be done by totally randomly moving a measurement device near a sample for a long time period, which is not suitable for practical operation. Although TFRs are independent of the input pattern in principle, protocols of the free-hand measurement are restricted by considering the practical operation. For example, measurement time within 10 s is favored for prompt identification of a target odor. As “randomly” is unclear and confusing, specific instructions on how to expose a measurement device to samples will be helpful to operators without professional expertise in sensor technologies. To popularize odor identification based on the free-hand measurements, optimization of measurement protocols under such restricted conditions is desired. The data analysis methods, especially the feature selection—the appropriate selection of signal features for the subsequent data analysis—of TFRs, also needs to be improved accordingly.

In this study, we have first developed a palm-size wireless measurement device equipped with MSS for the free-hand measurement (Figure 1). The aim of this study is to optimize the measurement protocol of the free-hand measurement and provide guidelines for developing accurate machine learning classification models from TFRs datasets. By using the wireless measurement device equipped with a 4-channel MSS chip, we measured vapors of four liquids (i.e., water, hexane, methanol, and acetone) through the free-hand measurement with different protocols. Datasets of TFRs were created from the measurement data and analyzed by clustering and machine learning classification. On the basis of the result of data analysis, we discuss the effect of measurement protocols and feature selection on the vapor identification.

## 2. Materials and Methods

### 2.1. MSS

An MSS is a silicon (Si)-based sensor fabricated through a MEMS process [16,17,18]. A thin membrane is suspended by four beams, in which piezoresistors (*R*_1_, *R*_2_, *R*_3_, and *R*_4_) are embedded. A receptor material, which absorbs gas species, is coated on the membrane. According to this unique structure, surface stress caused by the expansion/shrinkage of the receptor material accumulates in the four beams, resulting in resistance change of the piezoresistors. As the piezoresistors consists a full Wheatstone bridge circuit, the signal output (*V*_out_) is described as the following equation:(1)$$V_{\mathrm{out}}=\frac{V_{B}}{4}\left(\frac{\Delta R_{1}}{R_{1}}-\frac{\Delta R_{2}}{R_{2}}+\frac{\Delta R_{3}}{R_{3}}-\frac{\Delta R_{4}}{R_{4}}\right)$$
where *V*_B_ is the bridge voltage applied to the circuit. In this study, we used 4-channel MSS chips provided by NanoWorld AG (Neuchâtel, Switzerland).

Receptor materials were coated on MSS by inkjet spotting. In this study, we used four polymers as receptor materials: poly(4-vinylphenol), polyepichlorohydrin, poly(styrene-co-allyl alcohol), and poly(methyl methacrylate) for channel Ch.1, Ch.2, Ch.3, and Ch.4, respectively. The polymers were dissolved in N,N-dimethylformamide (DMF) at a concentration of 1.0 g/L. The solution was delivered onto each MSS channel with LaboJet-500SP (MICROJET Corporation, Shiojiri, Japan). The stage on which the MSS chip was set during the inkjet spotting was heated to 80 °C. Two hundred droplets were shot onto Ch.1 while 300 droplets were shot onto Ch.2, Ch.3, and Ch.4. The MSS chip coated with the 4 polymers were observed by an upright microscope (Eclipse Ni, Nikon Corporation, Tokyo, Japan).

The basic sensing properties of the MSS chip was evaluated by a gas-sensing measurement system equipped with two mass flow controllers (MFCs) which provided dried nitrogen. The same measurement system was used in our previous studies [22,23,24]. One MFC (MFC1) was connected to a glass vial which contains a sample liquid; dried nitrogen from MFC1 delivered the vapor of the sample liquid. The nitrogen almost saturated with the sample vapor was then diluted by dried nitrogen provided by the other MFC (MFC2). The diluted sample vapor was delivered to a sensor chamber in which the MSS chip was set. To evaluate the sensing properties of the MSS chip, sample vapors and dried nitrogen were alternately delivered to the MSS chip. First, the MSS chip was purged with dried nitrogen for 30 s (the gas flow rates of MFC1 and MFC2 were 0 and 100 standard cubic centimeters per minute (sccm), respectively). Then, a sample vapor was injected (the gas flow rates of MFC1 and MFC2 were 20 and 80 sccm, respectively) for 30 s, followed by the dried nitrogen purge for 30 s. This cycle of the sample vapor injection and the dried nitrogen purge was repeated 4 times. In this study, we used water, hexane, methanol, and acetone as sample liquids. Before each measurement, the MSS was “washed” by water vapor for approximately 60 s. The signal output was recorded at a sampling rate of 20 Hz with an analog-to-digital converter (NI-9214, NI, Austin, TX, USA). Bridge voltage *V*_B_ of −0.5 V was applied with a digital-to-analog converter module (NI-9269, NI, Austin, TX, USA). The gas flow of the two MFCs were controlled with a custom program based on LabVIEW (NI, Austin, TX, USA).

Hexane and DMF were purchased from Wako Pure Chemical Industries (Osaka, Japan) while methanol and acetone from Kanto Chemical Co., Inc. (Tokyo, Japan) and Sigma-Aldrich (St. Louis, MO, USA), respectively. Ultrapure water was prepared with Direct-Q UV3 (Merck & Co., Inc., Kenilworth, NJ, USA). All the polymers were purchased from Sigma-Aldrich (St. Louis, MO, USA).

### 2.2. Free-Hand Measurements

Gas sensing measurements were done with the wireless measurement device as shown in Figure 1. The dimensions of the device are 70 mm × 22 mm × 14 mm, and it weighs 20 g. Power was supplied by a lithium-ion battery. The 4-channel MSS chip was set at the top of the device. A bridge voltage of −1.0 V was applied to each channel. The signal output of each channel was recorded at a sampling rate of 20 Hz and 24-bit resolution, and the data were transmitted to a laptop by Bluetooth 4.2 (Bluetooth Low Energy). The power consumption of the device was less than 70 mW.

Free-hand measurements were performed on the 4 sample liquids in glass beakers: water, hexane, methanol, and acetone. The vapors were measured by manually bringing the wireless measurement device into the headspace of the beakers. In this study, three types of measurement protocols were performed; the device was brought into the headspace for 1, 3, and 5 times in 10 s. Hereafter, these protocols are described as “Protocol-1”, “Protocol-3”, and “Protocol-5”, (see Appendix A). Thirty measurement data were obtained for each liquid sample; the sample size of measurement data was 120 for each protocol. To avoid data bias caused by measurement conditions, measurement order for the samples was randomized. Furthermore, measurements were taken over more than two days for each measurement protocol. As the beginning and the end parts of the measurements tended to be disturbed, the time region from 2 to 9 s were used for analysis.

### 2.3. TFRs

A sensor is regarded as a control system in which gas concentration *c*(*t*) and a sensing signal *y*(*t*) are an input and an output, respectively. For a multichannel sensor system in which all the channels are assembled in close proximity, input *c*(*t*) is considered to be the same for each channel. The output signal of the *i*-th channel *y_i_*(*t*) can be described as a convolution of its transfer function *h_i_*(*t*) and an input *c*(*t*):(2)$$y_{i}\left(\tau \right)=\int_{0}^{\infty }h_{i}\left(\tau \right)c\left(t-\tau \right)d\tau$$
In the frequency domain, Equation (2) is written as the following formula:(3)$$Y_{i}\left(f\right)=H_{i}\left(f\right)C\left(f\right)$$
where *Y_i_*(*f*), *H_i_*(*f*), and *C*(*f*) refer to *y_i_*(*t*), *h_i_*(*t*), and *c*(*t*) in the frequency domain, respectively. As *C*(*f*) is the same for all the channels, the following equation holds between the *m*-th and the *n*-th channels.
(4)$$Y_{m}\left(f\right)=H_{m}\left(f\right)C\left(f\right)$$
(5)$$Y_{n}\left(f\right)=H_{n}\left(f\right)C\left(f\right)$$

Thus, the transfer function ratio (TFR) between the *m*-th and the *n*-th channels defined as *K_m,n_*(*f*) = *H_m_*(*f*)/*H_n_*(*f*) is represented as the following form:(6)$$K_{m,n}\left(f\right)=\frac{H_{m}\left(f\right)}{H_{n}\left(f\right)}=\frac{Y_{m}\left(f\right)}{Y_{n}\left(f\right)}$$

Gas species can be identified by *K_m,n_*(*f*) because *K_m,n_*(*f*) is specific to gas species. As *K_m,n_*(*f*) does not contain *C*(*f*), gas species can be identified only from sensing signals at each channel; controlling or monitoring the gas flow of a sample gas is not required.

In this study, TFRs were calculated from the time-series data from the 4 channels. To transform the time-domain data to the frequency domain, the fast Fourier transform (FFT) was applied to the time-series data. Before FFT, the mean value of time-series data was subtracted from the data, and Hanning window was applied. From the frequency-domain data, 6 TFRs were calculated according to Equation (6): namely, *K*_1,2_(*f*), *K*_1,3_(*f*), *K*_1,4_(*f*), *K*_2,3_(*f*), *K*_2,4_(*f*), *K*_3,4_(*f*). As the length of a time-series data is 8 s, the minimum unit of frequency is 0.125 Hz. According to the Nyquist–Shannon sampling theorem, the maximum frequency component of the data is 10 Hz, as the data were obtained at a sampling rate of 20 Hz. In this study, the offset value (0 Hz component) was not used. As TFRs are complex numbers, TFRs were divided into absolute values and arguments; logarithms of the absolute values were used as features because they vary in a wide range. Figure 2 shows an example of TFRs shown in graphs. Finally, the calculated TFRs (arguments and logarithms of absolute values) were concatenated to form 960-dimensional data for each measurement.

### 2.4. Cluster Analysis

Principal component analysis (PCA) was performed on the dataset of the TFRs to visualize the dataset. PCA is one of the most common dimensionality reduction algorithms—projecting the original data to new coordinates consisting of principal components. From a multidimensional dataset, PCA finds a direction that maximizes the variance of the dataset (PC1). The second principal component (PC2) is the direction that maximizes variance among all directions orthogonal to PC1. The further components are determined in the same manner.

To evaluate the quality of clusters, we employed the Davies–Bouldin (DB) index [25]. DB index is defined as the following equation:(7)$$DB=\frac{1}{n}\overset{n}{\underset{i}{\sum }}\underset{i\neq j}{\max}\left(\frac{\sigma_{i}+\sigma_{j}}{d\left(c_{i},c_{j}\right)}\right)$$
where *n* is the number of clusters. *c_i_* and *c_j_* are the centroids of cluster *i* and *j*, respectively, and *d*(*c_i_*, *c_j_*) is the distance between *c_i_* and *c_j_*. *σ_i_* and *σ_i_* denote the mean distance of all elements in cluster *i* and *j* from *c_i_* and *c_j_*, respectively. According to the definition, DB index becomes small for the clusters which are well-separated with each other.

### 2.5. Machine Learning Classification

We developed classification models based on logistic regression and random forests. For developing classification models based on logistic regression, TFRs datasets from the measurements were first standardized, and the dimensions were reduced by PCA. The logistic regression classifier was then trained with the datasets. In contrast, classification models based on random forests were developed without dimensionality reduction. The models were optimized and validated through a 5 × 5 nested cross-validation process. First, all the data were randomly divided into 5 subsets. One subset was used as a test dataset while the other 4 subsets were for a training dataset. A classification model was developed from the training dataset, and its performance was evaluated by the test dataset. By changing the combination of the 5 subsets for training and test datasets (“outer” cross-validation), both the mean accuracy and the standard deviation of classification models were evaluated. In the model building, the training dataset was further split into 5 subsets; one subset was for validation and the other 4 subsets were for model building. In this “inner” cross-validation, the hyperparameters of classification models were optimized by grid search. The hyperparameters of the models are summarized in Table 1. To develop classification models, scikit-learn was used in this study.

Note that this machine learning model is applicable only to the samples used in the training of the supervised learning. Unused samples or a mixture of used samples cannot be identified because the model classify any sample into the learned categories.

## 3. Results

### 3.1. Basic Properties of the MSS

Figure 3a shows the optical microscope image of the MSS chip. The receptor materials were coated on the 4 channels. The coatings did not deform or peel off from the membrane after the gas-sensing measurements with the gas flow line. Figure 3b–e shows the results of the gas-sensing measurements. Baseline offset values were subtracted from the signals. Each channel responded to the sample gases differently from each other.

Figure 4 shows examples of the sensing signals obtained through the free-hand measurement; the measurements were performed by Protocol-1, Protocol-3, and Protocol-5 for Figure 4a–c, respectively. Peaks can be clearly seen for each measurement protocol, reflecting the different concentration of the vapor inside and outside the beaker.

### 3.2. Cluster Analysis and Machine Learning Classification

The results of PCA on the dataset of TFRs are shown in Figure 5. Data were obtained by Protocol-1, Protocol-3, and Protocol-5 for Figure 5a–c, respectively. It is clear from Figure 5 that data points from the same samples form clusters on the PCA score plot, showing that the TFRs reflect interactions between receptor materials and sample vapors. However, the quality of cluster separation seems to differ between the three protocols; the cluster of water totally overlaps on that of hexane for Figure 5a, while they are separated from each other in Figure 5c. It seems that cluster separation improves as the number of exposures in a protocol increases. This trend can be quantitatively confirmed by focusing on the DB index. DB indices for the TFRs datasets obtained by Protocol-1, Protocol-3, and Protocol-5 are 4.87, 4.79 and 3.96, respectively. As a small DB index is expected for a dataset with good cluster separation, Protocol-5 achieved the best cluster quality among the three protocols.

The accuracies of the machine learning classification models are shown in Figure 6. As the models were developed by nested cross-validation, error bars (standard deviations) of the accuracies are also shown. For both two classifiers, vapors could be identified with a mean accuracy of more than 0.8, showing that classification accuracy of about 0.8 could be achieved with the free-hand measurement without further optimization in measurement protocols and feature selection. For the logistic regression classifier, the classification model built from the Protocol-3 dataset exhibited slightly higher accuracy than the one built from the Protocol-1 dataset. In contrast, accuracies of classification models based on random forests are not significantly different. The random forest-based model developed from Protocol-5 TFRs dataset exhibited the highest accuracy (0.96 ± 0.03).

### 3.3. Dependence on Frequency Range

As the protocols of the free-hand measurements affect the cluster quality of TFRs’ datasets and the accuracy of classification models, it is expected that there is an optimal frequency range in TFRs for vapor identification. To investigate such dependence on the frequency range, we prepared 4 TFRs datasets that contain different frequency components: 0.125–2.500 Hz (Data I), 2.625–5.000 Hz (Data II), 5.125–7.500 Hz (Data III), and 7.625–10.000 Hz (Data IV). Figure 7 shows the results of PCA on these datasets. In any protocol, the clusters collapse as the frequency region shifts to higher frequencies (left to right in Figure 7). Although the dependence of cluster separation on protocols is not clear from Figure 7, it seems that the dataset obtained by Protocol-5 shows better cluster separation than the ones obtained by Protocol-1 and Protocol-3. For example, in Data I (Figure 7a,e,i), three clusters (water, hexane, and acetone) overlap for Protocol-1 (Figure 7a) and Protocol-3 (Figure 6e) while only two clusters (water and acetone) overlap for Protocol-5 (Figure 7i) except two outliers. The DB indices for the datasets are summarized in Figure 8. As we have seen in Figure 7, the DB index increases as the frequency components contained in the datasets shift to higher frequencies; cluster quality deteriorates for TFRs’ datasets which contain only high-frequency components. This trend can be seen for all the protocols. In any frequency region, datasets from Protocol-5 exhibited better cluster quality than the ones from Protocol-1 and Protocol-3.

Classification models were also developed for each dataset. The accuracies of the classification models are summarized in Figure 9. Similar to the DB index, classification accuracy tends to deteriorate as the frequency range contained in a dataset becomes high. No significant difference is observed among the three protocols. Of all the classification models, the random forest-based classification model developed from Data I of the Protocol-5 TFRs dataset exhibited the highest classification accuracy (0.98 ± 0.03), which is slightly better than that of the random forest-based model built from all the frequency components of the Protocol-5 TFRs dataset (0.96 ± 0.03).

## 4. Discussion

On the basis of the results of the cluster analysis and the machine learning classification, it is evident that high-frequency components of TFRs do not represent the interaction between sample vapors and receptor materials. As such a high-frequency region contains more electrical noises than a low-frequency region does, using the high-frequency components as features is not effective for vapor identification. This is also confirmed from the statistical investigation of TFRs; for most of log|*K_m,n_*| and arg*K_m,n_*, high-frequency components exhibit higher standard deviation than low-frequency components (see Appendix A). Low-frequency components, on the other hand, seem to reflect the dynamics of the sensing mechanism. According to the viscoelastic model proposed by Wenzel et al. [26,27], the dynamics of nanomechanical sensors, including MSS, can be expressed by two time constants: gas diffusion constant *τ*_s_ and time constant for relaxation modulus *τ*_r_. As a rough estimation, the inverse of the time constants corresponds to the frequency components. Previous studies have shown that these time constants range from 1 to 100 s [27,28]; hence frequency components of 0.01 to 1 Hz correspond to the time constants. To precisely evaluate the TFRs of such a frequency range, the input to the sensor system (*c*(*t*) and *C*(*f*) in Equations (2) and (3), respectively) should contain sufficient frequency components of that range as well. To investigate the input frequency components that each protocol contains, we simulated the inputs *c*(*t*) of the protocols as shown in Figure 10a. The signals were simulated as binary waveforms; the values alternately took 0 and 1, which correspond to the vapor concentration *c*(*t*) outside and inside the beaker, respectively. The time interval of 0 and 1 domains was 8/3 (=2.67), 8/7 (=1.14), 8/11 (=0.72) s for Protocol-1, Protocol-3, Protocol-5, respectively. A waveform consists of 160 data points, which is equivalent to a measurement with a sampling rate of 20 Hz for 8 s. Figure 10b shows the result of FFT to the simulated waveforms; the absolute values of the frequency components of the waveforms are plotted for the protocols. In the frequency region below 2.5 Hz, the first peak shifts to a higher frequency as the number of waves in the input waveform increases. Another feature is that Protocol-5 covers more frequency components than Protocol-1 or Protocol-3 in the frequency region below 1 Hz; the sums of the absolute values of the frequency components in the range from 0.125 to 1 Hz are 98.54, 103.27, 116.96 for Protocol-1, Protocol-3, and Protocol-5, respectively. As we have discussed, a frequency range below 1 Hz contains more information of sensing dynamics than other ranges; protocols that cover the entire frequency components in that range are preferable for vapor identification based on TFRs. Thus, the TFRs dataset obtained by Protocol-5 exhibited the best cluster quality and achieved the most accurate classification model among the three TFRs datasets.

Although the difference in classification accuracies between the two classifiers are not significant, classification models based on random forests scored higher mean accuracies than the ones based on logistic regression for every dataset (except Data IV of Protocol-3). As the logistic regression classifier was trained by TFRs datasets of which the dimensionality was reduced by PCA, the learning was more contaminated by the noise components than the random forests classifier, which is a kind of ensemble learning algorithm. This is because PCA determines the principal components on the basis of variance, which is intrinsically not related to the importance of each feature. If classification models were developed without dimensionality reduction, training of logistic regression classifier would suffer from the “curse of dimensionality” [29], resulting in low accuracies. In contrast, random forest-based models can be trained with low bias and low variance owing to its learning procedure; a random forest classifier is developed by aggregating multiple decision tree models developed from data subsets sampled by bootstrapping, leading to high applicability to high dimensional datasets. Thus, a random forest is more preferable as a classifier for developing classification models from high-dimensional TFRs datasets than logistic regression.

Toward a practical implementation of machine olfaction based on the free-hand measurement, it is also important to evaluate the dependence on gas concentration. In principle, TFRs do not depend on the concentration of gases as long as the responses of the sensors are regarded to be linear. For a low concentration range, however, the classification accuracy would be worsened because the signal to noise ratio becomes lowered. To obtain effective sensing signals, the measurement device needs to be exposed to the samples for a longer time than for the case of headspace gases of the liquid samples—high concentration samples.

## 5. Conclusions

In this study, we have developed palm-size wireless odor measurement devices equipped with MSS chips for the free-hand measurement. Although the free-hand measurement is independent of the input pattern in principle, roughly fixed measurement protocols are favored from a viewpoint of practical use. To explore the optimal measurement protocol in a practical condition with the measurement time of 10 s, we have examined three types of measurement protocols that are easy to perform with the wireless measurement device and investigated their effects on odor identification. From the TFRs datasets created from the measurement data obtained with such protocols, features (i.e., frequency components) used for developing machine learning classification models were optimized. It has been clarified that high-frequency components of TFRs did not effectively contribute to vapor identification because of noise components, whereas classification models developed from TFRs datasets with low-frequency components yielded high accuracies. In addition, for accurate vapor identification, we have demonstrated that measurement should be performed in such a way that an MSS chip is exposed to the sample vapor several times in ten seconds. In such a measurement protocol, the input waveform—the time variation in concentration of a sample vapor—covers sufficient frequency components below 1 Hz, which are related to the sensing dynamics of MSS. The utilization of other receptor materials with quicker dynamics than the typical ones might be useful for collecting effective data from other frequency ranges, especially above 1 Hz. As the next step toward a practical application of this method, the repeatability between measurement devices needs to be evaluated. Considering the compactness and its usability, the present odor measurement device has great potential in mobile use and IoT applications. The present study will contribute to practical applications of machine olfaction and its implementation in society.

## Figures and Tables

**Figure 1 sensors-20-06190-f001:**
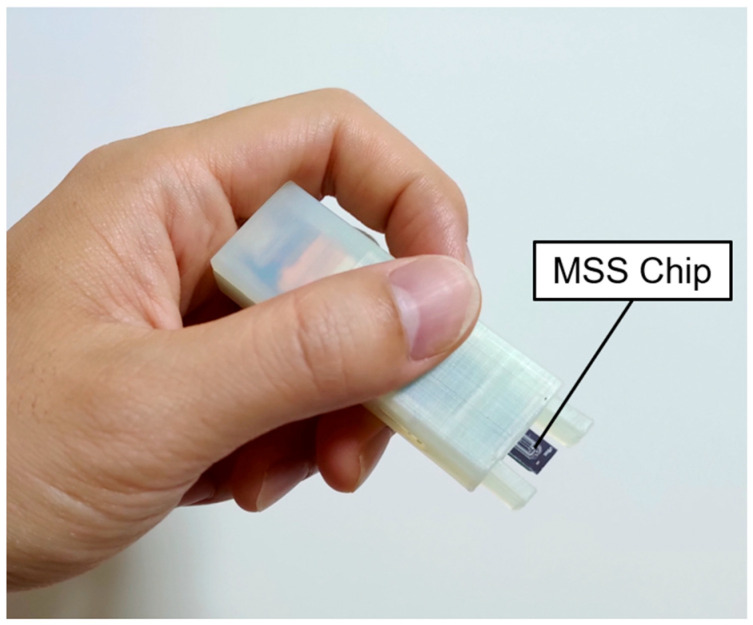
Picture of wireless measurement device for the free-hand measurement.

**Figure 2 sensors-20-06190-f002:**
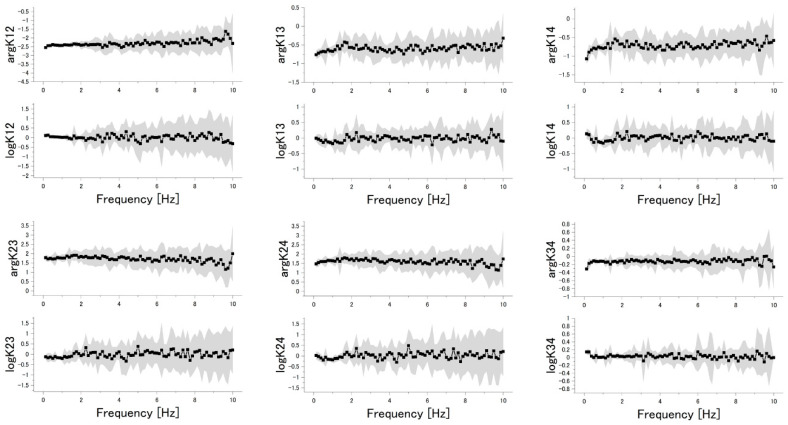
Transfer function ratios (TFRs) of the data obtained through Protocol-5. Mean values and standard deviations of log|*K_m,n_*| and arg*K_m,n_* are plotted as squares and shaded areas, respectively. Water was used as a sample.

**Figure 3 sensors-20-06190-f003:**
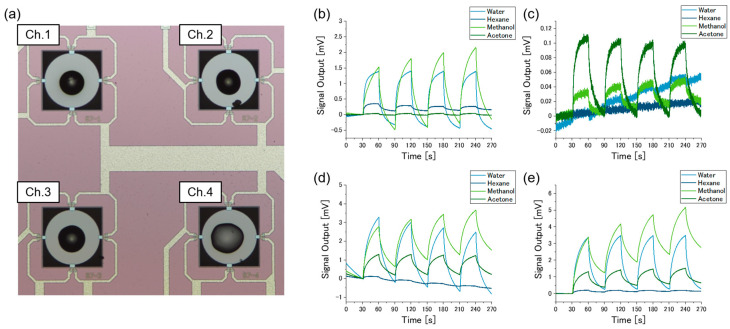
(**a**) Microscope image of the Membrane-type Surface stress Sensors (MSS) chip. Poly(4-vinylphenol), polyepichlorohydrin, poly(styrene-co-allyl alcohol), and poly(methyl methacrylate) were coated on Ch.1, Ch.2, Ch.3, and Ch.4, respectively. Results of the gas-sensing measurements for (**b**) Ch.1, (**c**) Ch.2, (**d**) Ch.3, and (**e**) Ch.4.

**Figure 4 sensors-20-06190-f004:**
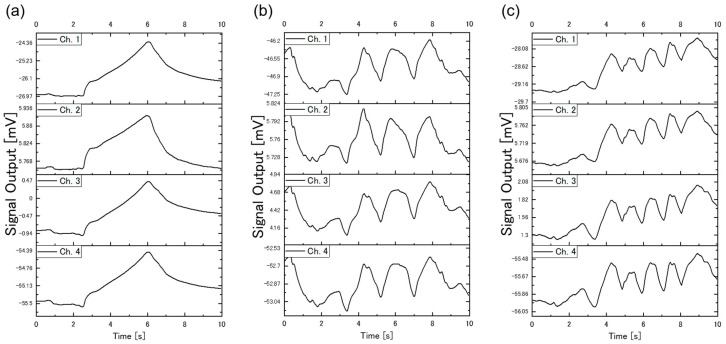
Examples of sensing signals for each protocol; the measurement device was brought into the headspace of the beaker (**a**) once (Protocol-1), (**b**) 3 times (Protocol-3), and (**c**) 5 times (Protocol-5). Water was used as a sample.

**Figure 5 sensors-20-06190-f005:**
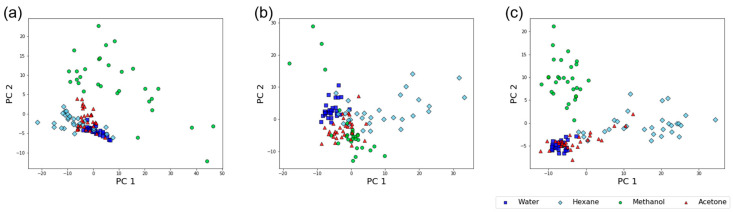
PCA score plots for TFRs datasets. Measurement data were obtained by (**a**) Protocol-1, (**b**) Protocol-3, and (**c**) Protocol-5.

**Figure 6 sensors-20-06190-f006:**
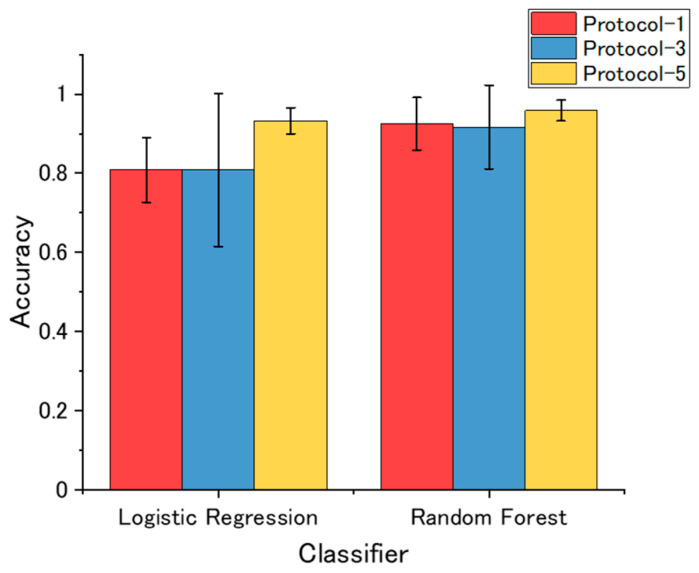
Summary of the classification models.

**Figure 7 sensors-20-06190-f007:**
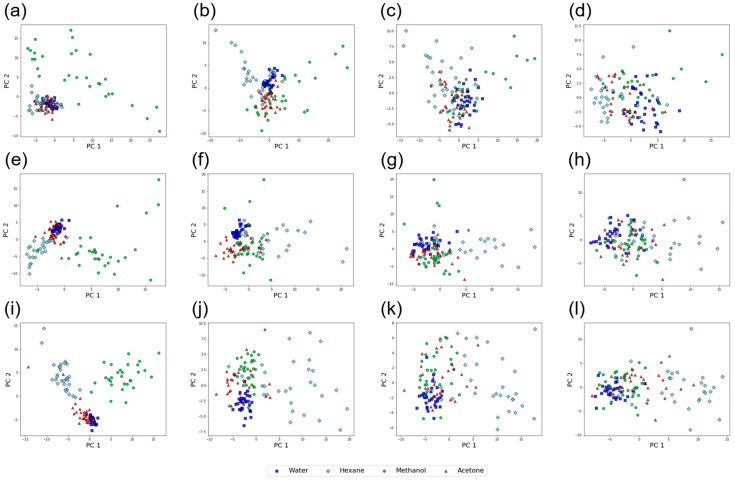
PCA score plots for TFRs datasets. (**a**)–(**d**) Data I, Data II, Data III, and Data IV from the TFRs datasets obtained by Protocol-1; (**e**)–(**h**) Data I, Data II, Data III, and Data IV from the TFRs datasets obtained by Protocol-3; (**i**)–(**l**) Data I, Data II, Data III, and Data IV from the TFRs datasets obtained by Protocol-5.

**Figure 8 sensors-20-06190-f008:**
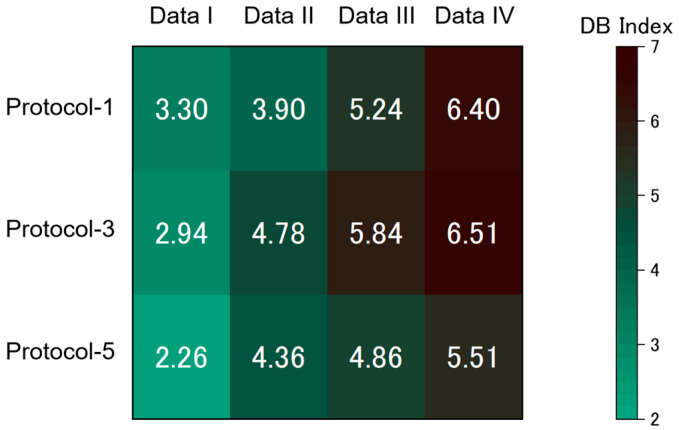
Colormap of Davies–Bouldin (DB) indices for the TFRs datasets.

**Figure 9 sensors-20-06190-f009:**
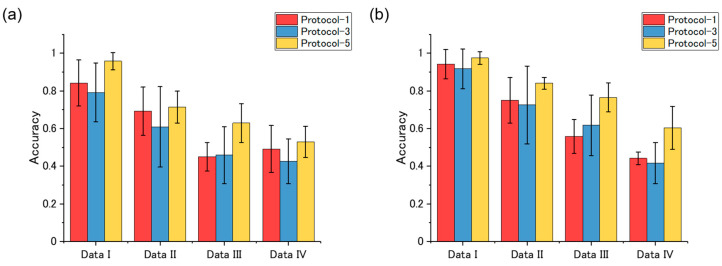
Summary of the classification models. Accuracies of classification models based on (**a**) logistic regression and (**b**) random forests.

**Figure 10 sensors-20-06190-f010:**
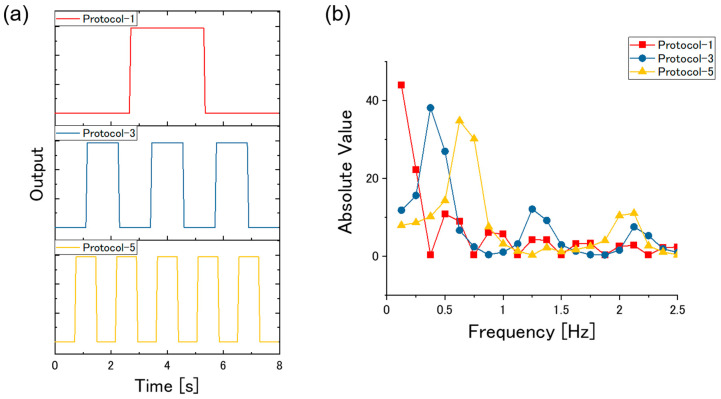
(**a**) Simulated waveform. (**b**) Absolute value of the frequency components for the simulated waveforms.

**Table 1 sensors-20-06190-t001:** Hyperparameters of the models.

	Classification Models Based on Logistic Regression	Classification Models Based on Random Forests
Hyperparameters (Name of the parameter in the scikit-learn library)	Number of principal component analysis (PCA) components (n_components) Regularization parameter (C)	Number of estimators (n_estimators)

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
