# Peer review of "Development of a Mobile Device for Odor Identification and Optimization of Its Measurement Protocol Based on the Free-Hand Measurement"

_sensors, 2020, doi:10.3390/s20216190_

Round 1

Reviewer 1 Report

The authors applied a ready made microelectromechanical system (MEMS) with polymer gas receptors for detection of sample odours. 

The work contains a good introduction to the field of study, a decent description of the sensing device and results. 

The Authors have conducted a number of experiments showing high odour detection performance by applying the proposed processing pipeline consisting of the sensor signals, the transfer function ratios (TFRs) and the applied machine learning algorithms. 

My critical comments of the submission are not major, and are the following: 

(1) the title promises too much, i.e. that the odour concentrations will be measured, whereas we are getting just odour detection device. Here, no specifications of odour concentrations that trigger the detections have been specified also. I would suggest the Authors might consider the title, e.g. "A mobile odour detection device and optimization of the detection protocol".  

(2) The Authors provide a number of plots of the signal outputs from the sensors. A more meaningful information, however, in my mind is hidden in the TFRs (transfer function ratios). Can you include plots for these transfer functions in the manuscript?

(3) It is claimed that "... the maximum frequency component of the data is 10 Hz". How this is ensured? The gas concentration dynamics is slow? 

(3) Equation (6) needs more explanation, I suggest to add an index 'j' to the DB symbol to indicate for which cluster we compute the index  

(4) Finally, it would be interesting to know what is the 'reatability' of the TFRs for different fabrication items of the sensors. It seems the Authors have just tested one sensor item. 

I suggest printing the paper after making the above corrections and suggestions

Author Response

Dear Reviewer 1,

We would gratefully acknowledge the reviewer for his/her critical reading of the manuscript, his/her interests in our work, and his/her essential comments. We believe that we have successfully revised our manuscript, which answers the reviewer’s questions. Please see the attached file.

Reviewer 2 Report

In this study, the authors used the wireless measurement device equipped with a 4-channel MSS (membrane-type stress sensors) chip to measure and identify vapors of four liquids (i.e. water, hexane, methanol, and acetone) through the free-hand measurement with different protocols. Datasets of TFRs (Transfer function ratios) were created from the measurement data and analyzed by clustering and machine learning classification, and the highest classification accuracy reaches 0.98±0.03. Sensor array manufacturing is not the focus of research. 4-channel MSS chips were purchased and 4 different receptor materials(polymer) were coated on MSS by inkjet spotting. surface stress caused by the expansion/shrinkage of the receptor material accumulates in the four beams, resulting in resistance change of the piezo resistors. As the piezo resistors consists a full Wheatstone bridge circuit, the signal output is different. Innovation of this study: 1. TFRs (transfer function ratios) are used to handle the dataset. Gas concentration is not contained in this method so gas species can be identified only from sensing signals at each channel, controlling or monitoring the gas flow of a sample gas is not required. 2. Optimize the measurement protocol of the free-hand measurement. An MSS chip should be exposed to the sample vapor several times in ten seconds. The input waveform covers sufficient frequency components below 1Hz are related to the sensing dynamics of MSS. 3. This measurement protocol could be applied to other practical close-to-gas-source odor recognition due to its compactness and usability. Problems: 1. Line 35 “Among the five human senses, olfaction is the only human sense that has not been implemented 36 in society as practical sensing applications.” Should be revised as “Among the five human senses: touch, sight, hearing, olfaction and taste, olfaction is the one of human senses that has not been implemented in society as practical sensing applications.” Reason: taste is also not implemented widely. 2. “An ideal free-hand measurement protocol should be done by totally randomly moving a measurement device near a sample for a long time period“(line 80) “To popularize odor identification based on the free-hand measurements, optimization of measurement protocols under such restricted conditions is desired.(line 87)” The measurement protocol optimization in subsequent articles mainly includes measurement frequency. But "restrictions" should be discussed further: a) The study uses headspace measurement to measure the signal——The sensor is placed on a beaker containing liquid droplets to measure the signal. Transfer function ratio eliminates the influence of gas concentration on gas type recognition, regarding gas concentration in 4 channels are the same. It should be further discussed whether the transfer function ratio changes under different concentrations (different distances from the gas source) and the effect on the accuracy of odor recognition. b) The influence of mixed gas sources on the accuracy of measurement results should be further discussed.

Author Response

Dear Reviewer 2,

We would gratefully acknowledge the reviewer for his/her critical reading of the manuscript, his/her interests in our work, and his/her essential comments. We believe that we have successfully revised our manuscript, which answers the reviewer’s questions. Please see the attached file.

Round 2

Reviewer 2 Report

All the comments have been replied.